# Impact of the Coronavirus Disease 2019 [COVID-19] Pandemic on Post-Acute Care of Patients with Heart Failure and the Effectiveness of Vaccine Prevention

**DOI:** 10.3390/healthcare12212171

**Published:** 2024-10-31

**Authors:** Lin-Yuan Chang, Chin-Yi Chao, Jin-Long Huang, Yun-Yu Chen, Chi-Yen Wang, Wen-Lieng Lee, Wei-Wen Lin

**Affiliations:** 1Cardiovascular Center, Taichung Veterans General Hospital, Taichung 407219, Taiwan; linda20322@yahoo.com.tw (L.-Y.C.); chinyichao@gmail.com (C.-Y.C.); golden@vghtc.gov.tw (J.-L.H.); r01847021@gmail.com (Y.-Y.C.); b8902092@vghtc.gov.tw (C.-Y.W.); wenlieng.lee@gmail.com (W.-L.L.); 2Nursing Department, Taichung Veterans General Hospital, Taichung 407219, Taiwan; 3Nursing Department, National Taichung University of Science and Technology, Taichung 40343, Taiwan; 4Cardiovascular Research Center, College of Medicine, National Chung Hsing University, Taichung 402202, Taiwan; 5Cardiovascular Research Center, National Yang Ming Chiao Tung University School of Medicine, Taipei 112304, Taiwan; 6Department of Medical Education, Taichung Veterans General Hospital, Taichung 40705, Taiwan; 7Department of Medical Research, Taichung Veterans General Hospital, Taichung 40705, Taiwan; 8Heart Rhythm Center, Taipei Veterans General Hospital, Taipei 112201, Taiwan; 9Department of Medicine, Chung-Shan Medical University, Taichung 40201, Taiwan; 10Life Science Department, Tunghai University, Taichung 407224, Taiwan

**Keywords:** heart failure, post-acute care, COVID-19, vaccine

## Abstract

Background: The Heart Failure Post-Acute Care [HF-PAC] program is a specialized healthcare program aimed at providing comprehensive care and support for patients with heart failure [HF] as they transition from acute hospital settings to home. But the impact of the coronavirus disease 2019 [COVID-19] pandemic on the HF-PAC program remains unknown. Furthermore, the effects of the comprehensive COVID-19 vaccination program on these patients with HF-PAC warrants further investigation. Methods: A total of 265 patients with acute decompensated HF were admitted to the hospital between May 2020 and October 2022. Of these, 159 patients underwent planned HF-PAC follow-up for 6 months, followed by scheduled follow-up visits every 3 months and unscheduled telephone randomized visits for at least another 6 months. Results: The program completion rate was nearly 92%. COVID-19 significantly impacted patients with HF-PAC, leading to an increased mortality [13.3%] compared to before the pandemic [6.5%]. In our patient cohort, 83% had received at least 1 dose of vaccine and 61% had received > 3 doses. Of these patients with HF-PAC, 34% contracted COVID-19 infection post discharge, and 8.8% died owing to the infection. Of the mortality group, 42.9% patients were not vaccinated, and 28.6% received 1 vaccine dose, and their vaccination rate was lower than in the survival group [*p* = 0.01]. Conclusions: The COVID-19 pandemic had a significant impact on patients enrolled in the HF-PAC program; receiving more than 3 doses of the COVID-19 vaccine was associated with a significant reduction in mortality rates among these patients.

## 1. Introduction

Patients with heart failure [HF] often have other risk factors for cardiovascular diseases, such as diabetes, hypertension, and chronic obstructive pulmonary disease, and may be at a higher risk of complications from coronavirus disease 2019 [COVID-19] [1,2,3]. Severe acute respiratory syndrome coronavirus 2 [SARS-CoV-2] primarily targets the respiratory system, leading to severe pneumonia; however, it can also affect other organs, such as the heart and kidneys. The purpose of this work is to evaluate the impact of the COVID-19 infection on patients with HF and the effectiveness of vaccine prevention. COVID-19 can directly damage the heart muscles [4], leading to myocarditis, HF exacerbations, and arrhythmias. The inflammatory response triggered by the virus can lead to a cytokine storm, further exacerbating cardiac heart disease [5]. The exact incidence of this condition is not well established, as it may vary depending on the country, population demographics, and availability of healthcare resources [6,7]. The Heart Failure Post-Acute Care [HF-PAC] program [8] is a specialized healthcare initiative designed to provide comprehensive care and support to patients with HF and facilitate the transition from acute care hospital settings to home. The new treatment model invites not only cardiologists, but also HF case managers, clinical pharmacists, cardiac surgeons, and social workers to provide integrated and comprehensive care as a multi-professional medical team. Thus, the National Health Insurance Administration in Taiwan launched an HF-PAC program in 2014 to provide integrated care for patients with HF during the early phases of recovery. This program has been shown to improve left ventricular function, quality of life, readmission, and mortality rates [9]. However, the impact of the COVID-19 pandemic on the HF-PAC program remains unknown. Comprehensive COVID-19 vaccination has effectively reduced mortality and morbidity in different disease populations; however, there are currently few studies discussing the effect of vaccination on patients with HF [10,11,12,13,14]. This study aims to assess risk factors and symptoms associated with higher mortality in HF patients infected with COVID-19. Additionally, we also evaluate the effectiveness of vaccination in this vulnerable population during the peak of the pandemic, from May 2020 to October 2022. We hypothesize that COVID-19 vaccination can effectively reduce mortality in patients with HF-PAC.

## 2. Methods

We used a retrospective cohort design because historical data are available and can be used to determine exposure and outcome status in patients with HF-PAC [15]. Patients hospitalized with acute decompensated heart failure [ADHF] during the peak of the COVID-19 pandemic [May 2020 to October 2022] were recruited from Taichung Veterans General Hospital, Taiwan, a central medical center [Appendix A]. Convenience sampling was used due to the availability of retrospective data during the specified period. A total of 159 patients who met the inclusion criteria for receiving HF-PAC prior to discharge were included in the analysis. Comprehensive clinical data were collected from existing medical records, including the most recent laboratory and echocardiographic findings within 1 year before the COVID-19 infection. In this program [Appendix B], the medical team educated the patients and their families, including on the treatment of symptoms of worsening HF, more dietary restrictions in daily life, compliance with medication, and consultations with nutritionists and rehabilitation physicians to assist in the management of their daily life. The recruited patients were assessed every 1 month after discharge and had a minimum follow-up period of 6 months. The program was approved by the Ethics Committee and Institutional Review Board [IRB, approval number CE23336B] of Taichung Veterans General Hospital, Taiwan [approval date 16 August 2023]. The committee waived the requirement for written informed consent owing to the retrospective cohort nature of the study.

### Statistical Analysis

Descriptive statistics were employed to analyze the variables. Normality was tested using the Kolmogorov–Smirnov test for sample sizes greater than 50 and the Shapiro–Wilk test for sample sizes of 50 or smaller. Continuous variables were presented as mean ± standard deviation for normally distributed data. Categorical variables were expressed as absolute numbers and percentages. Group comparisons for continuous variables were conducted using Student’s *t*-test for two groups and One-Way ANOVA for more than two groups. Categorical variables were compared using the chi-square test. The Cox proportional hazards regression model was employed to estimate hazard ratios [HRs] for outcomes, considering both univariable and multivariable effects. The cumulative event-free survival curve was adjusted after controlling for age, sex, systolic blood pressure, left ventricular ejection fraction, flu vaccination status, and the use of medications [TX2 and TX3]. Additionally, changes in LVEF before and after HF-PAC were analyzed using generalized estimating equations [GEEs] to estimate the beta coefficients, allowing for the evaluation of the effects over time. Statistical analyses were conducted using SPSS Statistics version 23 [IBM Corp., Armonk, NY, USA] [15]. A *p*-value < 0.05 was considered significant.

## 3. Results

A total of 265 patients with acute decompensated HF were admitted to the hospital between May 2020 and October 2022. Of these, 159 patients underwent a 6-month planned HF-PAC follow-up, then regular follow-up every 3 months, and random unscheduled telephone calls for at least another 6 months. The program had a completion rate of nearly 92% [Table 1]. The patients’ mean age was 65.9 ± 14.9 years, of whom 24.5% were women. Patient details included New York heart association functional class III [86.8%] and class IV [4%], and their mean left ventricular ejection fraction [LVEF] was 32 *±* 11%. Their mean systolic blood pressure [SBP] was 130.1 *±* 26.3 mmHg and diastolic blood pressure [DBP], 77.1 *±* 20.8 mmHg; they presented with chronic kidney disease stage 3–4 [41%], stage 5 [4.4%], hypertension [57.9%], diabetes mellitus [42.8%], and obesity [body mass index > 24; 62.9%]. Most patients [93.1%] were treated with renin–angiotensin–aldosterone system inhibitors, before the COVID-19 epidemic [Table 1]. During the HF-PAC follow-up period, 54 patients [34%] were infected with COVID-19 [40 patients were diagnosed with positive antibody tests and 14 patients with positive polymerase chain reaction [PCR] tests]. Of these, 43 patients were isolated at home, 5 patients were at the anti-epidemic hotel, and 6 at the hospital’s isolation intensive care unit. Fourteen patients [8.8%] died from COVID-19 infection. Compared to patients not infected or who survived COVID-19 infection, those patients who died from the infection were older, but the difference was not statistically significant [*p* = 0.074]. Patients with chronic obstructive pulmonary disease had a higher mortality rate but this was not significant [*p* = 0.066]. However, the SBP [116.5 *±* 17.8 mmHg], DBP [64.9 *±* 6.2 mmHg], and heart rates [70.2 *±* 14.1 beats/min] were significantly lower [all *p* < 0.05].

### 3.1. Lab Data and Electrolyte Profile

Patients with HF-PAC [Table 2] with high BUN [40.5 *±* 21.0 mg/dL], creatinine [2.18 *±* 0.99 mg/dL], and lactate [30.1 *±* 24.65 mg/dL] during hospitalization exhibited higher mortality if infected with COVID-19 after discharge from the hospital [*p* < 0.05]. Additionally, significantly more patients with mean measures for anemia [10.88 *±* 3.03 mg/dL], lactic acidosis [30.13 *±* 24.65 mg/dL], and hypocalcemia [7.98 *±* 0.87 mg/dL] were in the mortality group [*p* < 0.05]. The heart failure biomarkers including NT-ProBNP [11,693.12 *±* 10,149.78 pg/mL] and cardiac troponin T [cTnT] [167.54 *±* 248.47 ng/L] were also significantly higher [*p* < 0.05] in the mortality group compared to surviving patients, regardless of COVID-19 infection status.

### 3.2. Clinical Symptoms and Mortality

Among all the patients with HF and COVID-19 infection [Table 3], six [15%] patients exhibited no symptoms and were tested owing to exposure to other patients with COVID-19. All 14 patients who died of infection presented with symptoms. Dyspnea was significantly higher among patients who died [*p* < 0.001], but other upper respiratory tract infection symptoms [such as fever and productive cough] were not significantly different. There were no significant differences in nervous symptoms such as headache, dizziness, muscle pain, and bone pain. The incidence of gastrointestinal symptoms was low in both the living [11.8%] and deceased groups [21.4%].

### 3.3. LVEF and Survival Rate

For patients who were either uninfected with or surviving COVID-19 infection, their LVEF improved compared to baseline at the start of the HF-PAC follow-up [*p* = 0.053, Figure 1, Table 4]. In contrast, the LVEF of patients who died from COVID-19 did not significantly improve [*p* < 0.001], despite receiving guideline-directed medical treatment [GDMT] [Figure 1, Table 4].

### 3.4. Medication

More than 90% of patients received HF neprilysin, including angiotensin-converting-enzyme inhibitor or angiotensin receptor blocker or angiotensin receptor/neprilysin inhibitor therapy [ACEi/ARB/ARNi] and beta-blockers [Table 1]. In the mortality group, fewer patients received ACEis/ARBs/ARNis [78.6%] and beta-blockers [71.4%] when they were discharged because of lower blood pressure and heart rate, but this difference was not statistically significant. Three types of medical treatments, including nonsteroidal anti-inflammatory drugs [NSAIDs], antiviral drugs [molnupiravir and Paxlovid], and traditional Chinese medicine [National Research Institute of Chinese Medicine 101, NRICM101], were used in patients with COVID-19 [Table 4], and 88.5% patients received a single medication. Nearly one-third of the patients took NRICM101, whereas one-third of the patients took antiviral drugs, with no significant differences between the groups.

### 3.5. Vaccine

There are four vaccines authorized for use in Taiwan: Pfizer–BioNTech, Oxford–AstraZeneca, Moderna, and Medigen Vaccine Biologics Corporation COVID-19 vaccines. Among the patients, 132 [83%] received > 1 dose of vaccine 28 days before HF-PAC follow-up [Table 5]. Of these, 69 [66%] patients who were not infected with COVID-19 and 26 [65%] patients who survived the infection received ≥ 3 doses of the vaccine. In contrast, six [42.9%] patients who died after the infection did not receive the vaccine, and four [28.6%] received only 1 dose of vaccine. Patients who received > 2 doses of the vaccine had a significantly higher survival rate than those who did not. [*p* = 0.01]. Most patients had received pneumococcal and influenza vaccinations. Thus, in the patient cohort, 83% had received at least 1 dose of vaccine and 66% had received > 3 doses. However, 44 [34%] patients had the COVID-19 infection after discharge, and 14 [8.8%] patients died owing to the infection. In the mortality group, 42.9% patients were unvaccinated and 28.6% had received 1 dose of vaccine, and the vaccination rate was significantly lower than in the survival group [*p* = 0.01].

### 3.6. Survival Curve

Patients with HF-PAC and COVID-19 infection who received >3 doses of the vaccine had a 100% one-year survival rate and a 79.6% two-year survival rate [Figure 2, Table 6]. For unvaccinated patients, the one-year survival rate was 40%, and all died after 15 months of follow-up [*p* < 0.001].

## 4. Discussion

The HF-PAC program in Taiwan targets hospitalized patients with ADHF [LVEF < 40%]. These participants received 6 months of PAC in a comprehensive HF clinic, which improved their HF functional class, physical activity, quality of life, and readmission rates [16]; the average 1-year mortality rate was 6.5%. This health insurance-reimbursed HF-PAC program reduces healthcare costs for patients discharged after hospitalization for HF [17]. In our patient cohort, several factors influenced the outcomes of patients with HF-PAC and COVID-19 after hospital discharge.

### 4.1. Heart Rate [HR], BP, Electrolyte, and NT-proBNP

Under beta-blocker treatment, patients with ADHF and a heart rate of <90/min at discharge experienced a significantly lower 1-year mortality rate [18]. Hypotension is often associated with poor prognosis in patients with ADHF [19,20] and may indicate severe cardiac dysfunction. The HR is regulated by the autonomic nervous system and reflects the metabolic demands of the body. Chronotropic incompetence [CI] is defined as the inability to increase HR during stress to match metabolic demands [21]. Hypotension and slow HR are important factors limiting the titration of HF treatments in routine practice. During hospitalization for acute decompensated HF, patients with lower HR, systolic BP, and diastolic BP, which indicated a poor general condition, had a higher mortality rate after COVID-19 infection.

Renal function impairment and imbalances in electrolytes such as sodium, potassium, and calcium can significantly affect the prognosis and management of HF [22,23]. In patients with HF-PAC, various serum electrolytes and biochemical markers are regularly followed up and corrected, if necessary. Serum calcium is involved in the excitation–contraction coupling of cardiomyocytes. Abnormal calcium regulation leads to delayed cell repolarization and increases the risk of fatal arrhythmias [24]. Calcium levels were significantly lower in the mortality group than in the survivors [Calcium: 7.98 ± 0.87 mg/dL, *p* < 0.005]. While we could not confirm arrhythmias as the cause of death, low calcium levels warrant attention to prevent potential arrhythmias. Remote monitoring using wearable ECG devices may be an effective approach for this group of patients [25]. Anemia reduces oxygen-carrying capacity, increases cardiac workload, and worsens heart function over time [26]. In patients with HF-PAC, anemia was a significant indicator of poor prognosis after COVID-19 infection [*p* = 0.019]. Our data suggest that correcting anemia may be crucial for optimizing HF management and improving overall outcomes. Mitchell et al. [27] reported about 60% of patients lost their sense of smell and taste with COVID-19 infection. These symptoms are related to the severity of the disease and improve in most patients during the recovery period. Sensory dysfunction is a common symptom in patients, even after full vaccination. However, in our patient cohort with HF-PAC, most were fully vaccinated, and only one patient exhibited sensory abnormalities. This discrepancy may be owing to regional or ethnic differences.

### 4.2. LVEF and Mortality

The LVEF is an important prognostic indicator in patients with HF. The worse the LVEF, the higher the likely mortality rate [28,29,30]. Of the HF-PAC patients hospitalized for HF, 90% presented with systolic HF [LVEF < 40%]. Post GDMT, most patients exhibited symptomatic and LVEF improvement before discharge. However, patients whose LVEF does not improve after GDMT require meticulous monitoring during outpatient follow-up, as their mortality rate may be elevated if diagnosed with COVID-19 infection. This may be because there is little viable myocardium left in the heart, and medications cannot improve this function. In these patients, persistently elevated NT-proBNP and cTnT levels after GDMT indicate a higher mortality rate after COVID-19 infection [31,32,33].

### 4.3. COVID-19 Medication

Frail patients with multiple comorbidities are often discharged with low BP and HRs, limiting the use of comprehensive HF medications. As a result, the usage rate of ARBs/ACEis/ARNis and beta-blockers was relatively low in this group [Table 1]. Although the mortality rate was higher in these patients upon COVID-19 infection, the small sample size limited the statistical significance. Notably, some studies suggest that ARB/ACEi usage does not increase mortality in patients with HF as compared with non-users.

Once patients with HF-PAC were infected with COVID-19, nearly 50% of the patients treated their symptoms with NSAIDs, such as acetaminophen or ibuprofen, for symptom relief. Of the patients, 30% used antiviral drugs, including Paxlovid [nirmatrelvir/ritonavir] and molnupiravir. Notably, nearly 30% also used traditional Chinese medicine treatment. NRICM101 is a novel traditional Chinese medicine developed by the NRICM to prevent and treat COVID-19 [34]. It is manufactured as a herbal drug powder by pharmaceutical companies with good manufacturing practices to meet the needs of the global pandemic with a non-exclusive license. Basic and preliminary clinical studies have shown that it can effectively inhibit viral proliferation, reduce lung inflammation, and improve patient symptoms. However, further clinical studies are required to confirm its efficacy.

### 4.4. Vaccinations and Mortality in Patients with HF-PAC

Alvarez-Garcia et al. reported a high incidence of mechanical ventilation [22.8%] and mortality [40.0%] in patients with HF who were hospitalized for COVID-19 at the beginning of the pandemic, but none of these patients were vaccinated [35].

Sindet-Pedersen et al. reported the safety of vaccination in patients with HF, with no increased risk of worsening HF, myocarditis, thromboembolism, and mortality [36]. Only a few studies have reported the effectiveness and survival rate of vaccination in patients with HF. The Health Promotion Administration of Taiwan approved vaccination from March 2021, prioritizing patients with HF. In this small, single-center study [Figure 2, Table 6], our data strongly suggest that patients with HF who received the COVID-19 vaccine may effectively increase their survival rates. These patients did not experience any significant adverse effects, such as myocarditis or thromboembolism, during follow-up after receiving the COVID-19 vaccine [37]. In retrospective cohort studies of patients with HF who received the COVID-19 vaccination, fully vaccinated patients had lower mortality rates than unvaccinated or partially vaccinated individuals [fewer than 2 doses] [38,39]. These studies showed that COVID-19 vaccination can effectively reduce the risk of respiratory failure, intensive care unit admission, and mortality in patients with HF. Our data are consistent with these studies and further demonstrate that patients with HF-PAC who were able to receive > 3 doses of the vaccine before discharge had significantly lower mortality rates than patients not fully vaccinated. While the Taiwanese government subsidizes influenza vaccination for people aged > 65 years old, pneumococcal vaccination is self-funded, resulting in about 70% of the patients in our study receiving the influenza vaccine and only 20% receiving pneumococcal vaccine. There was no difference in the mortality between these two groups of patients.

### 4.5. Study Limitations

This study had certain limitations. First, only a small number of patients from a single medical center were observed for 6 months. Therefore, we did not observe the possibility of reinfection or long COVID symptoms. Second, the patients were selected by the HF-PAC team and deemed suitable for active treatment and regular follow-up. Patients who could not cooperate or had serious conditions were excluded. If these excluded patients became infected, they may have had higher mortality rates. Third, the tracking of these patients relied only on telephone calls and questionnaires. In the future, if telemetry monitoring is used, it may reflect the actual conditions of these patients in a more timely manner.

## 5. Conclusions

The HF-PAC program provides multidisciplinary team care for patients with HF after hospital discharge, reducing readmissions and mortality. However, the COVID-19 pandemic significantly impacted this patient group. Patients discharged with low BP, slow HR, poor renal function, anemia, or persistently high NT-ProBNP levels face increased mortality if infected with COVID-19. However, COVID-19 vaccination is highly beneficial for this group, with >3 doses effectively reducing the mortality rate. In future research directions, we will specifically focus on the post-infection symptoms of patients with HF-PAC, determining the incidence of long COVID-19 conditions [40], and providing pharmacological treatment, when necessary. Furthermore, we will closely monitor the impact of viral mutations on these patients, as mutations may reduce vaccine efficacy, worsen disease severity, and affect the accuracy of existing diagnostic tools.

## Figures and Tables

**Figure 1 healthcare-12-02171-f001:**
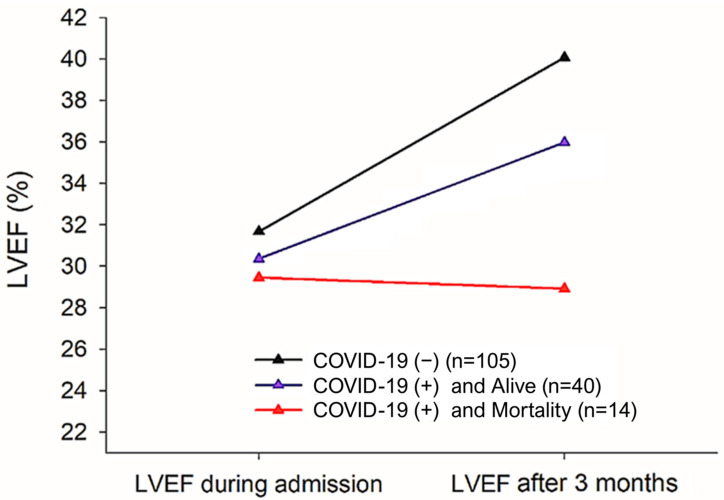
LVEF change before and after heart failure-post acute care. Following 3 months of drug treatment post discharge, patients with HF [COVID-19 [−], N = 105 and COVID-19 [+] and survival, N = 40] exhibited improvements in LVEF as observed during echocardiography follow-up. Drug treatment had no significant effect on a small number of patients [COVID-19 [+] and mortality, N = 14], and the LVEF did not increase; once these patients developed COVID-19, the mortality rate was higher [*p* < 0.001]. LVEF: left ventricular ejection fraction; COVID-19: coronavirus disease 2019; GEE, generalized estimating equation.

**Figure 2 healthcare-12-02171-f002:**
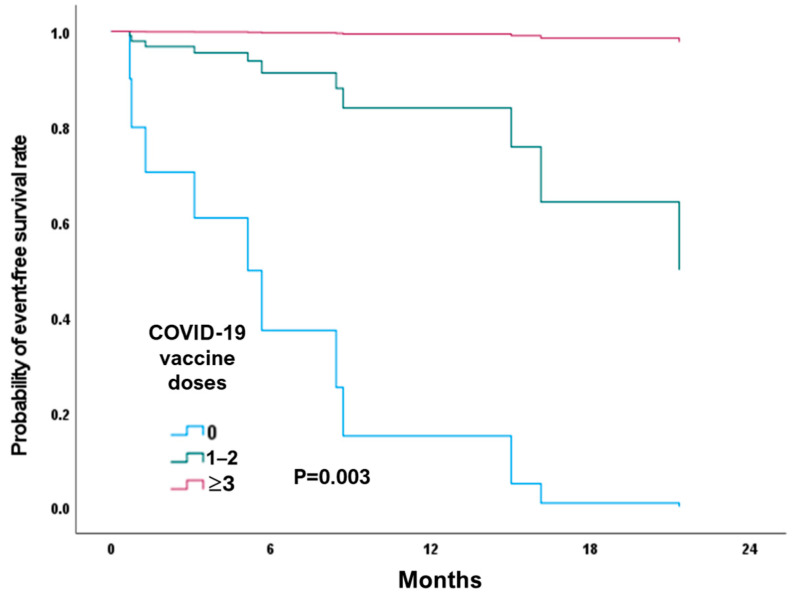
Kaplan–Meier curve comparison of survival according to COVID-19 vaccine doses in patients with heart failure. Patients with acute decompensated heart failure who did not receive vaccinations succumbed within 18 months of COVID-19 infection. Higher survival rates were found among patients who received 3 or more doses of vaccine, even if they contracted COVID-19 [*p* = 0.003].

**Table 1 healthcare-12-02171-t001:** Baseline characteristics among patients with heart failure who received post-acute care.

Variables	COVID-19 [−] [*n* = 105]	COVID-19 [+] and Alive [*n* = 40]	COVID-19 [+] and Mortality [*n* = 14]	*p*-Value
Age	63.88 ± 14.86	62 ± 13.95	72 ± 13.13	0.07
Sex				0.61
Male	79 [75.2%]	29 [72.5%]	12 [85.7%]	
Female	26 [24.8%]	11 [27.5%]	2 [14.3%]	
BMI	26.71 ± 5.19	26.29 ± 5.17	24 ± 3.96	0.23
Education level				0.27
Elementary school	27 [25.7%]	10 [25%]	6 [42.9%]	
High school	48 [45.7%]	14 [35%]	2 [14.3%]	
College and university	30 [28.6%]	16 [40%]	6 [42.9%]	
Marriage				0.87
Unmarried	10 [9.5%]	2 [5%]	1 [7.1%]	
Married	82 [78.1%]	34 [85%]	12 [85.7%]	
Divorced	4 [3.8%]	1 [2.5%]	1 [7.1%]	
Widowed	9 [8.6%]	3 [7.5%]	0 [0%]	
NYHA functional class heart failure				0.84
Fc 2	6 [5.7%]	1 [2.5%]	0 [0%]	
Fc 3	93 [88.6%]	38 [95%]	14 [100%]	
Fc 4	6 [5.7%]	1 [2.5%]	0 [0%]	
Length of hospital stay [day]	6.69 ± 5.05	5.88 ± 4.8	10.93 ± 9.4	0.04
Comorbidities, *n* [%]				
Hypertension	57 [54.3%]	26 [65%]	9 [64.3%]	0.44
Diabetes mellitus	42 [40%]	18 [45%]	8 [57.1%]	0.45
Overweight [BMI > 25]	67 [63.8%]	25 [62.5%]	8 [57.1%]	0.89
Hyperlipidemia	20 [19%]	9 [22.5%]	3 [21.4%]	0.89
Coronary artery disease	49 [46.7%]	23 [57.5%]	9 [64.3%]	0.29
Chronic obstructive pulmonary disease	8 [7.6%]	3 [7.5%]	4 [28.6%]	0.07
Arrhythmia	22 [21%]	9 [22.5%]	5 [35.7%]	0.46
Cigarette smoking	52 [49.5%]	21 [52.5%]	9 [64.3%]	0.58
Clinical presentation				
SBP [mmHg]	134.13 ± 26.34	129.18 ± 24.04	116.5 ± 17.77	0.03
DBP [mmHg]	81.09 ± 20.83	78.38 ± 15.77	64.86 ± 6.24	0.002
Heart rate [beats/min]	86.47 ± 21.71	78.45 ± 16.99	70.21 ± 14.1	0.01
Respiratory rate [rpm]	18.71 ± 2.7	18.98 ± 2.35	18.93 ± 2.06	0.68
Laboratory data				
Chronic kidney disease, CKD stage				0.40
Stage 1	6 [5.7%]	5 [12.5%]	1 [7.1%]	
Stage 2	52 [49.5%]	17 [42.5%]	4 [28.6%]	
Stage 3	27 [25.7%]	13 [32.5%]	5 [35.7%]	
Stage 4	16 [15.2%]	3 [7.5%]	4 [28.6%]	
Stage 5	4 [3.8%]	2 [5%]	0 [0%]	
Prescription HF medications, *n* [%]				
ARB/ACEi/ARNi				0.08
Beta-blockers	95 [90.5%]	37 [92.5%]	10 [71.4%]	0.09
MRA	72 [68.6%]	34 [85%]	10 [71.4%]	0.14
Ivabradine	13 [12.4%]	6 [15%]	3 [21.4%]	0.63
SGLT2i	16 [15.2%]	10 [25%]	2 [14.3%]	0.36
Echocardiography				
LVEF [before admission]	0.32 ± 0.11	0.3 ± 0.07	0.29 ± 0.08	0.84
LVEF [at discharge from hospital]	0.4 ± 0.1	0.36 ± 0.11	0.29 ± 0.11	0.003

Between May 2020 and October 2022, during the peak of the COVID-19 epidemic in Taiwan, 169 patients with heart failure received post-acute care at the hospital. Of these, 54 patients were infected with COVID-19 after being discharged from hospital, and 14 subsequently died from the infection. Patients with low systolic and diastolic blood pressure, bradycardia, and low LVEF at discharge had a higher mortality rate once infected during follow-up. COVID-19: Coronavirus disease 2019; NYHA: New York Heart Association; Fc: functional class; BMI: body mass index; BP: blood pressure; ACEi: angiotensin-converting-enzyme inhibitors; ARB: angiotensin receptor blockers; ARNi: angiotensin receptor/neprilysin inhibitors; MRA: mineralocorticoid receptor antagonist; SGLT2i: sodium/glucose cotransporter 2 inhibitor; LVEF: left ventricular ejection fraction.

**Table 2 healthcare-12-02171-t002:** Serum laboratory data at admission in patients with heart failure.

Variables	COVID-19 [−] [n = 105]	COVID-19 [+] and Alive [n = 40]	COVID-19 [+] and Mortality [n = 14]	*p*-Value
Glutamate oxaloacetate transaminase [GOT] U/L	68.73 ± 191.99	53.93 ± 98.69	67.23 ± 121.07	0.99
Glutamate pyruvate transaminase [GPT] U/L	43.83 ± 78.00	52.14 ± 116.63	42.71 ± 66.35	0.64
Albumin [g/dL]	3.51 ± 0.66	3.79 ± 0.64	3.37 ± 0.54	0.15
Total bilirubin [mg/dL]	1 ± 1.17	0.68 ± 0.40	0.93 ± 0.48	0.38
Direct bilirubin [mg/dL]	0.59 ± 0.89	0.42 ± 0.41	0.38 ± 0.26	0.87
Prothrombin time [second]	11.67 ± 2.95	13.3 ± 5.57	14.14 ± 4.98	0.02
Blood urea nitrogen [BUN, mg/dL]	30.51 ± 22.89	30.86 ± 17.02	40.5 ± 20.96	0.05
Creatinine [mg/dL]	1.56 ± 1.03	1.98 ± 1.96	2.18 ± 0.99	0.007
White blood cell [WBC, 109/L]	8443.27 ± 3107.76	8073.13 ± 2556.36	7657.14 ± 3896.57	0.65
Hemoglobin [g/dL].	13.2 ± 2.43	12.95 ± 2.30	10.88 ± 3.03	0.02
Platelet [109/unit]	222.68 ± 81.46	193.63 ± 71.34	190.14 ± 101.61	0.09
Sodium [Na, mEq/L]	138.63 ± 4.07	138.11 ± 3.34	139.14 ± 4.83	0.61
Potassium [K, mEq/L]	4.07 ± 0.55	4.18 ± 0.59	4.44 ± 0.71	0.08
Chloride [Cl, mEq/L]	102.35 ± 4.82	101.17 ± 4.45	102.44 ± 8.25	0.79
Calcium [Ca, mg/dL]	8.57 ± 0.59	8.94 ± 0.57	7.98 ± 0.87	0.005
Magnesium [Mg, mg/dL]	2.21 ± 0.39	2.25 ± 0.30	2.5 ± 0.91	0.64
Phosphorus [P, mg/dL]	4.02 ± 0.99	3.92 ± 0.87	4.07 ± 1.99	0.83
High-sensitivity C-reactive protein [hs-CRP, mg/dL]	3.47 ± 4.42	3.9 ± 6.31	5.21 ± 7.66	0.97
Lactate [mg/dL]	11.72 ± 9.12	15.01 ± 5.09	30.13 ± 24.65	0.005
Total cholesterol [mg/dL]	158.35 ± 40.66	158.18 ± 37.83	156 ± 63.41	0.66
High-density lipoprotein cholesterol [HDL-C, mg/dL]	43.08 ± 12.70	49.91 ± 13.17	54.71 ± 21.66	0.08
Low-density lipoprotein cholesterol [LDL-C, mg/dL]	94.77 ± 34.09	78.85 ± 25.04	84.38 ± 49.33	0.11
Triglycerides [Tg, mg/dL]	128.82 ± 82.07	186.35 ± 371.13	97.22 ± 64.15	0.21
Creatine kinase [CK, U/L]	200.97 ± 439.23	99.56 ± 59.48	99.23 ± 51.61	0.70
Creatine kinase myocardial band isoenzyme [CK-MB, U/L]	14.02 ± 33.95	12.69 ± 12.20	7.92 ± 3.58	0.56
Troponin T [ng/L]	224.11 ± 762.06	49.56 ± 50.87	167.54 ± 248.47	0.02
N-terminal pro–B-type natriuretic peptide [NT-proBNP, pg/mL]	5824.52 ± 7668.09	4253.6 ± 6135.03	11,693.12 ± 10,149.78	0.006
Thyroid-stimulating hormone [TSH, mU/L]	2.48 ± 2.58	6.96 ± 8.48	3.7 ± 7.67	0.21
Free thyroxine [free T4, ng/dL]	1.05 ± 0.22	0.96 ± 0.19	1.27 ± 0.96	0.61
Hemoglobin A1c [HbA1c, %]	6.3 ± 1.34	6.35 ± 1.30	6.42 ± 2.42	0.50

**Table 3 healthcare-12-02171-t003:** Symptoms of survival or death due to COVID-19 in patients with HF.

	Alive [*n* = 40]	Mortality [*n* = 14]	*p*-Value
Symptoms associated with COVID-19 infection	34 [85%]	14 [100%]	0.32
URI symptoms	31 [91.18%]	13 [92.86%]	0.67
Dry cough	28 [90.32%]	10 [76.92%]	0.34
Dyspnea	5 [16.13%]	12 [92.31%]	<0.001 **
Fever	9 [29.03%]	4 [30.77%]	1
Sore throat	6 [19.35%]	1 [7.69%]	0.65
Productive cough	3 [9.68%]	1 [7.69%]	1
Pneumonia	0 [0%]	1 [7.69%]	0.3
Nervous symptoms	9 [26.47%]	3 [21.43%]	1
Dizziness	4 [44.44%]	1 [7.69%]	1
Muscle pain	3 [33.33%]	0 [0%]	0.51
Headache	2 [22.22%]	0 [0%]	1
Bone pain	1 [11.11%]	0 [0%]	1
Gastrointestinal symptoms	4 [11.76%]	3 [21.43%]	0.4
Smell and taste loss	1 [25%]	0 [0%]	1
Abdominal pain	1 [25%]	0 [0%]	1
Nausea	1 [25%]	1 [7.69%]	1
Diarrhea	1 [25%]	3 [21.43%]	0.14

URI: upper respiratory infection symptoms [including runny nose, sneezing, and throat pain]. Continuous data are expressed as mean ± SD. Categorical data are expressed as number and percentage. Using the Mann–Whitney U test, Chi-square test, and Fisher’s exact test, ** *p* < 0.01. Dyspnea [*n* = 12] was significantly higher among patients who died [*p* < 0.001]. There were no significant differences in upper respiratory tract infection symptoms and neurological symptoms. Only 1 patient experienced loss of smell and taste after COVID-19 infection, but there were no significant differences between the two groups.

**Table 4 healthcare-12-02171-t004:** Association between LVEF changes and survival from COVID-19 infection.

Group	LVEF Change: Beta (95% CI)	*p*-Value
COVID (−) (*n* = 105)	Ref = 1	
COVID (+) and Survival (*n* = 40)	−4.1 (−8.2–0.10)	0.053
COVID (+) and Mortality (*n* = 14)	−11.1 (−17.7–−4.60)	<0.001

LVEF: left ventricular ejection fraction; COVID-19: coronavirus disease 2019; GEE, generalized estimating equation.

**Table 5 healthcare-12-02171-t005:** Effects of different vaccine types and doses on patient survival or mortality after COVID-19.

	COVID-19 [−] and Alive [*n* = 105]	COVID-19 [+] and Alive [*n* = 40]	COVID-19 [+] and Mortality [*n* = 14]	*p*-Value
COVID-19 vaccine				0.01
no	18 [17%]	3 [7.5%]	6 [42.86%]	
1 dose	5 [5%]	5 [12.5%]	4 [28.57%]	
2 doses	13 [12%]	6 [15%]	2 [14.29%]	
3 doses or more	69 [66%]	26 [65%]	2 [14.29%]	
Pneumococcus vaccine				
PCV13	15 [14%]	6 [15%]	3 [21.43%]	0.78
PPV23	18 [18%]	11 [27.5%]	3 [21.43%]	0.38
Influenza vaccine	50 [48%]	25 [62.5%]	11 [78.57%]	0.04
COVID-19 medications				
NSAIDs		17 [48.57%]	6 [54.55%]	0.73
NRICM101		10 [28.57%]	3 [27.27%]	1
Antiviral drugs		12 [34.29%]	4 [36.36%]	1
Number of COVID-19 treatment types				
One type		31 [88.57%]	9 [81.82%]	0.62
Two types		4 [11.43%]	2 [18.18%]	1

Among the 159 patients, 132 [83%] received > 1 dose of vaccine 28 days before HF-PAC follow-up. Of these, 26 [65%] patients who survived the infection received ≥ 3 doses of the vaccine. In contrast, six [42.9%] patients who died after the infection did not receive the vaccine, and four [28.6%] received only 1 dose of vaccine. Patients who received > 2 doses of the vaccine had a significantly higher survival rate than those who did not. [*p* = 0.01]. COVID-19: coronavirus disease 2019; PCV13: pneumococcal conjugate vaccine against 13 types of pneumococcal bacteria; PPV23: pneumococcal polysaccharide vaccine protecting against 23 types of bacteria; NSAIDs: non-steroidal anti-inflammatory drugs; NRICM101: National Research Institute of Chinese Medicine 101.

**Table 6 healthcare-12-02171-t006:** Number of patients at risk.

Number of COVID-19 Vaccine Doses	Baseline	6 M	12 M	18 M	24 M
0	9	3	1	0	0
1–2	17	13	9	4	2
≥ 3	28	25	18	12	3

## Data Availability

The data presented in this study are available on request from the corresponding author.

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
