# Peer review of "Impact of the Coronavirus Disease 2019 [COVID-19] Pandemic on Post-Acute Care of Patients with Heart Failure and the Effectiveness of Vaccine Prevention"

_healthcare, 2024, doi:10.3390/healthcare12212171_

Round 1
Reviewer 1 Report
Comments and Suggestions for Authors
Analyzing the effect of the COVID-19 pandemic on The Heart Failure Post-Acute Care (HF-PAC) program through a May 2020- October 2022 study of 265 patients is the focus of this report.
The study is well-conceived, well-written, well-analyzed, and insightful. The weaknesses are not following the journal guidelines sufficiently.
Following the Instructions for Authors: https://www.mdpi.com/journal/healthcare/instructions, please make these changes.
Abstracts are to be no more than 250 words. This Abstract is 349. Please reduce the length of the Abstract to 250 words.
Citations are to be in square brackets, not round. Please change all the citations to square brackets.
The Introduction “should define the purpose of the work and its significance, including specific hypotheses being tested”, and “highlight the main conclusions”.
Please redo tables that are not in the required style. Table 4 is the most obvious example of a table requiring redoing.
Line by line suggested edits
58 Please provide the most recent research regarding the Heart Failure Post-Acute Care (HF-PAC) program to support citation 8.
73 Please provide the research question and indicate in what way this research is novel.
92 In the text, please explain the selection of descriptive statistics to analyze the variables.
95 Please check if “Student's” is the intended word. If so, describe this and provide a current citation.
103 Please explain in the text the selection of SPSS version 23 and provide a citation to COVID-19-related research using the same version of SPSS.
155 Table 3—the meaning of the first row is unclear. “Symptom” of what? It is also unclear whether “Symptoms” relates to the “Upper Respiratory Tract Infection” row or it is the heading for the symptoms to follow. Please improve Table 3.
167-168 Figure 1 is a figure and a table. Please separate the table from the Figure and rename and number it as a table.
215-216 Figure 2 is a figure and a table. Please separate the table from the Figure and rename and number it as a table.
217-221 The current Figure 2 has two captions. Figures must have one caption.
236 Please provide the most recent research regarding Chronotropic incompetence to support citation 20.
320 Please include future research suggestions.
321-329 Please move the limitations to a subsection of the Discussion.
Author Response
- Abstracts are to be no more than 250 words. This Abstract is 349. Please reduce the length of the Abstract to 250 words.
Answer 1: Thank you for your suggestion, we reduce the length to 243 words.
- Citations are to be in square brackets, not round. Please change all the citations to square brackets.
Answer 2: We have changed all the citations to square brackets.
- The Introduction “should define the purpose of the work and its significance, including specific hypotheses being tested”, and “highlight the main conclusions”.
Answer 3: In line 45, we add our purpose of the work” The purpose of the work is to evaluate the impact of the COVID-19 infection on patients with HF and effectiveness of vaccine prevention.”. In line 63-67, we add the significance and hypothesis of the work and highlight the main conclusion ” This study aims to assess risk factors and symptoms associated with higher mortality in HF patients infected with COVID-19. Additionally, we also evaluate the effectiveness of vaccination in this vulnerable population during the peak of the pandemic, from May 2020 to October 2022. We hypothesize that COVID-19 vaccination can effectively reduce mortality in patients with HF-PAC.”
- Please redo tables that are not in the required style. Table 4 is the most obvious example of a table requiring redoing.
Answer 4: Thank you for your suggestion, we have redo Table 3 and Table 4
Line by line suggested edits
- 58 Please provide the most recent research regarding the Heart Failure Post-Acute Care (HF-PAC) program to support citation 8.
Answer 5: We cite the most recent research in citation 9; Eur J Heart Fail. 2024 Jan;26(1):1-
- 73 Please provide the research question and indicate in what way this research is novel.
Answer 6: In line 63-67, we rewrite the main purpose and clinical significance of this work
- 92 In the text, please explain the selection of descriptive statistics to analyze the variables.
Answer 7: Thank you for your valuable feedback. We have revised the manuscript to clarify the selection of descriptive statistics in the statistical analysis section (Lines 90-97).
Statistical analysis
Descriptive statistics were employed to analyze the variables. Normality was test-ed using the Kolmogorov-Smirnov test for sample sizes greater than 50 and the Shapiro-Wilk test for sample sizes 50 or less. Continuous variables were presented as mean ± standard deviation for normally distributed data. Categorical variables were expressed as absolute numbers and percentages. Group comparisons for continuous variables were conducted using the Student’s t-test for two groups and One-way ANOVA for more than two groups. Categorical variables were compared using the Chi-square test.
- 95 Please check if “Student's” is the intended word. If so, describe this and provide a current citation.
Answer 8: In line 96, we rewrite this word “ Student’s “
- 103 Please explain in the text the selection of SPSS version 23 and provide a citation to COVID-19-related research using the same version of SPSS.
Answer 9: Researchers used SPSS v23 to explore trends in COVID-19 infections, hospitalization rates, and the efficacy of health interventions.
Reference:
Answer 9: Thank you for your feedback. We used SPSS version 23 for data analysis because it provides robust statistical tools suitable for our study design, including the Generalized Estimating Equations (GEE) model and Cox proportional hazards regression analysis. SPSS 23 has been widely utilized in various fields of research, including COVID-19-related studies, as it offers reliable performance for complex statistical methods. The use of SPSS 23 in COVID-19-related research has been documented, such as in the study by Marwah et al. (2024) examining the impact of the COVID-19 pandemic on antenatal care in India.
Reference:
Marwah S, Sharma P, Tripathi S, Arora D, Agarwal A, Malik A, Gaikwad H, Bajaj B. Effect of COVID-19 Pandemic on Knowledge, Attitude and Practices Towards Antenatal Care Among Antenatal Women: A Study From a Tertiary Care Hospital in Delhi, India. Cureus. 2024 Jan 20;16(1)
- 155 Table 3—the meaning of the first row is unclear. “Symptom” of what? It is also unclear whether “Symptoms” relates to the “Upper Respiratory Tract Infection” row or it is the heading for the symptoms to follow. Please improve Table 3.
Answer 10 : Thank you for your suggestion, we redo table 3, rewrite symptoms to “Symptoms associated with COVID-19 infection” and add explanation of URI symptoms.
- 217-221 The current Figure 2 has two captions. Figures must have one caption.
Answer 11: We remove one caption
- 236 Please provide the most recent research regarding Chronotropic incompetence to support citation 20.
Answer 12: We renew the most recent paper in citation 20 “Circ Res. 2024 May 10;134(10): 1348-1378.”
- 320 Please include future research suggestions.
Answer 13: We add in line 322: In future research directions, we will specifically focus on the post-infection symptoms of patients with HF-PAC, determining the incidence of long COVID-19 condition, and providing pharmacological treatment when necessary. Furthermore, we will closely monitor the impact of viral mutations on these patients, as mutations may reduce vaccine efficacy, worsen disease severity, and affect the accuracy of existing diagnostic tools.
- 321-329 Please move the limitations to a subsection of the Discussion.
Answer 14: We move the study limitations to a new subsection in line 328
Reviewer 2 Report
Comments and Suggestions for Authors
Impact of the Coronavirus Disease 2019 (COVID-19) Pandemic on Post-Acute Care of Patients with Heart Failure and Effectiveness of Vaccine Prevention
Overall, this manuscript is technically sound and well presented. The components of the research are represented appropriately, as regard the title, introduction, Results, discussion, and conclusion. Only minor comments are needed to be cleared especially in methods section before it can be considered for publications.
Comments that need addition, clarifications and corrections:
· Line 74-89: Mention the type of study design: the authors should describe in details the type of study design (e.g cohort study both retrospective e.g Comprehensive clinical data of every 86 patient with HF-PAC were collected from existing medical records and the most recent 87 laboratory and echocardiographic findings were obtained within 1 year before COVID-19 88 infection. and prospective e.g. The recruited patients were assessed every 1 month after discharge and had a minimum follow-up period of 6 months.
· Demonstrate type of sampling and sample size.
· What about the normality test in statistical analysis?
Author Response
Dear Reviewer 2,
Question 1 Line 74-89: Mention the type of study design: the authors should describe in details the type of study design (e.g cohort study both retrospective e.g Comprehensive clinical data of every 86 patient with HF-PAC were collected from existing medical records and the most recent 87 laboratory and echocardiographic findings were obtained within 1 year before COVID-19 88 infection. and prospective e.g. The recruited patients were assessed every 1 month after discharge and had a minimum follow-up period of 6 months.
Answer 1: Thank you for your insightful feedback. We have clarified the type of study design in the manuscript (Lines 70-89) to ensure greater transparency.
The study employed a retrospective cohort design. Patients hospitalized with acute decompensated heart failure (ADHF) during the peak of the COVID-19 pandemic (May 2020 to October 2022) were recruited from Taichung Veterans General Hospital, Taiwan, a central medical center. Convenience sampling was used due to the availability of retrospective data during the specified period. A total of 159 patients who met the inclusion criteria for receiving HF-PAC (Heart Failure Post-Acute Care) prior to dis-charge were included in the analysis. Comprehensive clinical data were collected from existing medical records, including the most recent laboratory and echocardiographic findings within 1 year before the COVID-19 infection. In this program [Appendix B] the medical team educated the patients and their families, including the treatment of symptoms of worsening HF, more dietary restrictions in daily life, compliance with medication, and consultations with nutritionists and rehabilitation physicians to assist in the management of their daily life. Comprehensive clinical data of every patient with HF-PAC were collected from existing medical records and the most recent labor-atory and echocardiographic findings were obtained within 1 year before COVID-19 infection. The recruited patients were assessed every 1 month after discharge and had a minimum follow-up period of 6 months. The program was approved by the Ethics Committee and Institutional Review Board [IRB, Approval number CE23336B] of Tai-chung Veterans General Hospital, Taiwan [Approval Date: August 16, 2023]. The com-mittee waived the requirement for written informed consent owing to the retrospective cohort nature of the study.
Question 2 Demonstrate type of sampling and sample size.
Answer 2: The study utilized a retrospective cohort design. Patients hospitalized with acute decompensated heart failure (ADHF) during the COVID-19 pandemic (from May 2020 to October 2022) at Taichung Veterans General Hospital, Taiwan, were included. Convenience sampling was applied, based on the availability of medical records during this period. A total of 159 patients who met the inclusion criteria for receiving Heart Failure Post-Acute Care (HF-PAC) before discharge were analyzed.
Question 3 What about the normality test in statistical analysis?
Answer 3: Thank you for your valuable feedback. We have revised the manuscript to clarify the application of normality testing in the statistical analysis section (Lines 90-97).
Statistical analysis
Descriptive statistics were employed to analyze the variables. Normality was test-ed using the Kolmogorov-Smirnov test for sample sizes greater than 50 and the Shapiro-Wilk test for sample sizes 50 or less. Continuous variables were presented as mean ± standard deviation for normally distributed data. Categorical variables were expressed as absolute numbers and percentages. Group comparisons for continuous variables were conducted using the Student’s t-test for two groups and One-way ANOVA for more than two groups. Categorical variables were compared using the Chi-square test.
Reviewer 3 Report
Comments and Suggestions for Authors
Dear Authors,
I would like to thank you for your efforts.
I have some comments that I wish it may help:
1- Your study duration from May 2020-October 2022, during this period there are more than 3 variants of virus (Wuhan, Alpha, Beta, may be Delta), which means different disease pathogenesis and severity.
Could you please breakdown your data based on viral variants?
2- Vaccinated group of patient received different origins of vaccines (Pfizer, .....etc), Could you please breakdown groups based on the vaccine and compare results ?
3- How do you justify that the significance of results is primarily derived from Heart failure post-acute care , not from chronic obstructive pulmonary disease or other anonymous factors?
Thank you
Author Response
Dear Reviewer 3.
Question 1. 1- Your study duration from May 2020-October 2022, during this period there are more than 3 variants of virus (Wuhan, Alpha, Beta, may be Delta), which means different disease pathogenesis and severity. Could you please breakdown your data based on viral variants?
Answer 1. Thank you for your question. In page 3, line 118. Among the 54 confirmed patients, 40 patients used commercially available serological testing reagents to confirm the infection. Only 14 patients were diagnosed using PCR testing, some of these patients were diagnosed in other hospitals. We learned of the infection during a post-event phone interview, so it is difficult for us to provide the variants type of virus that infected
Question 2. Vaccinated group of patient received different origins of vaccines (Pfizer, .....etc), Could you please breakdown groups based on the vaccine and compare results ?
Answer 2. We have statistics on the brands of vaccines used, and we are happy to provide them for your reference. However, due to the small number of patients, there is no statistical significance. However, because some vaccines receive different government subsidies, we did not list them in the article out of fear of inconvenience and necessary political considerations.
Question 3. How do you justify that the significance of results is primarily derived from Heart failure post-acute care , not from chronic obstructive pulmonary disease or other anonymous factors?
Answer 3. These patients have different comorbidities, which we list in Table 1. When a patient comes to the emergency room because of heart failure symptoms, specialist nurses and cardiologists will make an assessment based on the HF-PAC inclusion terms (Appendix A). Only those patients who meet the criteria will receive cardiology services. (Appendix B)
Round 2
Reviewer 1 Report
Comments and Suggestions for Authors
Thank you to the authors for the changes made. All have improved the work. Some changes are still required.
Line by line suggested edits.
52 Please find a supporting citation for 8 of research published since 2020. Please inform the reader if there has been no research on this program since 2015.
70 Please explain in the text the selection of a retrospective cohort design and provide a citation to similar current research using the same design.
91 Please explain the selection of descriptive statistics to analyze the variables.
104 Please cite similar research published since 2020 using version 23 of SPSS.
168-176 Please separate Figure 1 into a figure and a table and renumber the tables accordingly, referring to the newly numbered table in the text.
218-223 Please separate Figure 2 into a figure and a table and renumber the tables accordingly, referring to the newly numbered table in the text.
328-337 Please move the Limitations to subsection 4.5. of the Discussion.
Author Response
Question 1: 52 Please find a supporting citation for 8 of research published since 2020. Please inform the reader if there has been no research on this program since 2015.
Answer 1: Thank you for your valuable feedback. We have replaced reference 8 with the latest paper (Eur J Heart Fail. 2024 Jan;26[1] :1-4.)
Question2: 70 Please explain in the text the selection of a retrospective cohort design and provide a citation to similar current research using the same design.
Answer 2: We explain why we use a retrospective cohort design and provide a citation.
In Line 70-71, “We used a retrospective cohort design because historical data are available and can be used to determine exposure and outcome status in patients with HF-PAC (15)”
Alkhaneen H, Alsadoun D, Almojel L, Alotaibi A, Akkam A. Differences of Lipid Profile Among Ischemic and Hemorrhagic Stroke Patients in a Tertiary Hospital in Riyadh, Saudi Arabia: A Retrospective Cohort Study. Cureus. 2022 May 31;14(5):e25540. doi: 10.7759/cureus.25540. PMID: 35800812; PMCID:
Question 3: 91 Please explain the selection of descriptive statistics to analyze the variables.
Answer 3: Thank you for your valuable feedback. We have revised the manuscript to clarify the selection of descriptive statistics in the statistical analysis section (Lines 92-99).
Statistical analysis
Descriptive statistics were employed to analyze the variables. Normality was tested using the Kolmogorov-Smirnov test for sample sizes greater than 50 and the Shapiro-Wilk test for sample sizes 50 or less. Continuous variables were presented as mean ± standard deviation for normally distributed data. Categorical variables were expressed as absolute numbers and percentages. Group comparisons for continuous variables were conducted using the Student’s t-test for two groups and One-way ANOVA for more than two groups. Categorical variables were compared using the Chi-square test.
Question 4: 104 Please cite similar research published since 2020 using version 23 of SPSS.
Answer 4: We cite a similar paper using SPSS version 23, and add reference in line 107
Question 5: 168-176 Please separate Figure 1 into a figure and a table and renumber the tables accordingly, referring to the newly numbered table in the text.
Answer 5: We have separate Figure1 into Figure 1 and Table 4 in line 172-181
Question 6: 218-223 Please separate Figure 2 into a figure and a table and renumber the tables accordingly, referring to the newly numbered table in the text.
Answer 6: We have separate Figure2 into Figure 2 and Table 6 in line 225-231
Question 7: 328-337 Please move the Limitations to subsection 4.5. of the Discussion.
Answer 7: We move study limitations to subsection 4.5 in line 325
Reviewer 3 Report
Comments and Suggestions for Authors
Dear Authors,
I would like to thank you for your explanations and corrections.
I think that it is ready to go now.
Thank you.
Author Response
Thank you for your valuable advice